# Iron Deficiency and Restless Sleep/Wake Behaviors in Neurodevelopmental Disorders and Mental Health Conditions

**DOI:** 10.3390/nu16183064

**Published:** 2024-09-11

**Authors:** Osman S. Ipsiroglu, Parveer K. Pandher, Olivia Hill, Scout McWilliams, Melissa Braschel, Katherine Edwards, Robin Friedlander, Elizabeth Keys, Calvin Kuo, Marion Suzanne Lewis, Anamaria Richardson, Alexandra L. Wagner, David Wensley

**Affiliations:** 1H-Behaviours Research Lab, BC Children’s Hospital Research Institute, University of British Columbia, Vancouver, BC V5Z 4H4, Canadascout.mcwilliams@bcchr.ca (S.M.); 2Sleep/Wake-Behaviour Clinic, Sleep Program BC Children’s Hospital, Department of Pediatrics, University of British Columbia, Vancouver, BC V6H 3N1, Canada; 3Divisions of Developmental Pediatrics, Child and Adolescent Psychiatry, and Respirology, BC Children’s Hospital, Department of Pediatrics, University of British Columbia, Vancouver, BC V6H 3N1, Canada; 4Clinical Research Support Unit, BC Children’s Hospital Research Institute, Vancouver, BC V5Z 4H4, Canada; 5Division of Child and Adolescent Psychiatry, BC Children’s Hospital, Department of Psychiatry, University of British Columbia, Vancouver, BC V6H 3N1, Canada; 6School of Nursing, University of British Columbia (Okanagan), Kelowna, BC V1V 1V7, Canada; 7School of Biomedical Engineering, University of British Columbia, Vancouver, BC V6T 1Z4, Canada; 8Department of Medical Genetics, Faculty of Medicine, University of British Columbia, Vancouver, BC V6T 1Z4, Canada; 9BC Children’s Hospital Research Institute, University of British Columbia, Vancouver, BC V6T 1Z4, Canada; 10Pacific Autism Family Network, Richmond, BC V7B 1C7, Canada; 11Granville Pediatrics, 205-5678 Granville Street, Vancouver, BC V6M 1X7, Canada; 12Department of Pediatrics, Division of Neurology, Charité University Hospital Berlin, 10117 Berlin, Germany; 13Division of Respirology, BC Children’s Hospital, Department of Pediatrics, University of British Columbia, Vancouver, BC V6H 3N1, Canada

**Keywords:** attention deficit/hyperactivity disorder, autism spectrum disorder, circadian rhythm sleep disorders, fetal alcohol spectrum disorder, hematologic indices, insomnia, iron deficiency, prenatal alcohol exposure, restless leg syndrome, restless sleep disorder, sleep-disordered breathing

## Abstract

Iron deficiency (ID) and restlessness are associated with sleep/wake-disorders (e.g., restless legs syndrome (RLS)) and neurodevelopmental disorders (attention deficit/hyperactivity and autism spectrum disorders (ADHD; ASD)). However, a standardized approach to assessing ID and restlessness is missing. We reviewed iron status and family sleep/ID history data collected at a sleep/wake behavior clinic under a quality improvement/quality assurance project. Restlessness was explored through patient and parental narratives and a ‘suggested clinical immobilization test’. Of 199 patients, 94% had ID, with 43% having a family history of ID. ADHD (46%) and ASD (45%) were common conditions, along with chronic insomnia (61%), sleep-disordered breathing (50%), and parasomnias (22%). In unadjusted analysis, a family history of ID increased the odds (95% CI) of familial RLS (OR: 5.98, *p* = 0.0002, [2.35–15.2]), insomnia/DIMS (OR: 3.44, *p* = 0.0084, [1.37–8.64]), and RLS (OR: 7.00, *p* = 0.01, [1.49–32.93]) in patients with ADHD, and of insomnia/DIMS (OR: 4.77, *p* = 0.0014, [1.82–12.5]), RLS/PLMS (OR: 5.83, *p* = 0.009, [1.54–22.1]), RLS (OR: 4.05, *p* = 0.01, [1.33–12.3]), and familial RLS (OR: 2.82, *p* = 0.02, [1.17–6.81]) in patients with ASD. ID and restlessness were characteristics of ADHD and ASD, and a family history of ID increased the risk of sleep/wake-disorders. These findings highlight the need to integrate comprehensive blood work and family history to capture ID in children and adolescents with restless behaviors.

## 1. Introduction

Iron, classified as a trace element, is present in all of the body’s cells and serves as a component of hemoglobin and myoglobin. Its primary role involves carrying oxygen in both the bloodstream and muscles [1]. Recently, iron has received increasing attention due to its role in sleep disorders [2,3], as well as wake behaviors associated with mental health (MH) and/or neurodevelopmental disorders (NDDs) such as attention deficit hyperactivity disorder (ADHD) [4,5], autism spectrum disorder (ASD), and prenatal alcohol exposure/fetal alcohol spectrum disorder (PAE/FASD) [6].

During development, iron plays a substantial role in several central nervous system processes, including neurogenesis, the differentiation of brain cells, myelination, and the neurotransmitter metabolism. Perinatal iron deficiency (ID) in animal models may alter the neurochemical profile of the hippocampus [7], lead to dysfunction in the striatum accompanied by behavioral changes [8], and have persistent effects on brain myelination [9]. These proposed mechanisms are derived from iron’s modulating role in the brain and spinal cord [10], leading us to consider central ID rather than focusing exclusively on brain ID [11,12].

Iron is present as a trace element in neurons, oligodendrocytes, astrocytes, and microglia, and plays an essential role in the transfer of electrons and the synthesis of neurotransmitters such as dopamine, epinephrine, norepinephrine, and serotonin. These neurotransmitters are involved in various functions, including emotions, the sleep/wake cycle, movement, attention, memory, and learning through its cofactor function for phenylalanine hydroxylase [13] and tryptophan hydroxylase [14]. Furthermore, iron serves as an important cofactor for tyrosine hydroxylase which regulates the dopamine synthesis pathway by converting tyrosine to dihydroxy-phenylalanine (DOPA) [15,16]. Collectively, these findings suggest that both perinatal and postnatal ID might play a substantial role in behavioral medicine, including both sleep and wake behaviors.

Non-anemic ID is associated with multiple sleep/wake disorders; however, a standardized assessment of iron status in the diagnostic work-up for sleep/wake-disorders has not been established in clinical practice [3]. The most frequent causes of sleep/wake disturbances associated with central ID are sleep/wake disorders presenting with hyper-motor restlessness, hyper-arousability, restless leg syndrome (RLS) [17,18,19,20], periodic limb movements in sleep (PLMS) [21,22], and restless sleep disorder (RSD) [23]. RLS is a neurological sensorimotor disorder which is a frequently unrecognized and/or underestimated cause of intractable chronic insomnia [24,25]. The characteristic “urge-to-move” can manifest in several different body parts and is associated with paresthesias, discomfort, or even pain. As RLS was first described in adults and has been approached from an adult neurology perspective [26], the natural history of pediatric RLS has not been thoroughly investigated and pediatric RLS phenotypes have not yet been described. The role of RLS in neurodevelopmental disorders such as PAE/FASD [27], Down Syndrome [28], or ASD [29] has been questioned, but not investigated systematically. Diagnoses are made based on adult sleep medicine and reporting an “urge-to-move”, which presents as restlessness. This is alleviated by voluntary movements while awake, but results in disturbances in the onset and maintenance of sleep. PLMS are believed to be a continuum of RLS symptoms during sleep that impact sleep quality and are characterized by repetitive small movements of the toes, leading to larger movements involving the ankle and knee [21,22].

RLS, PLMS, and ADHD share hypermotor restlessness and hyper-arousability as common factors that lead to non-restorative sleep. While RLS may present with ADHD-like behaviors during the daytime [18,30], ADHD itself has been associated with a variety of sleep phenotypes [31] and is considered to be a 24 h disorder. RSD is characterized by frequent body movements of large muscle groups that disrupt both sleep and wakefulness, and was first described in 2018. Characterizing hyper-motor restlessness during sleep as a new diagnostic entity supports our understanding that disorders presenting with hypermotor-restlessness and/or hyper-arousability need an in-depth phenotyping approach from a pediatric sleep medicine perspective [23].

Given the experimental and clinical evidence of the effects of iron on the central nervous system and sleep/wake behaviors [2,3,12,32,33,34,35], we implemented an iron status assessment as a standard investigation in our clinical setting. Here, we present findings on iron status in children and adolescents referred to a Sleep/Wake-Behaviour Clinic at a tertiary care center with a focus on the common neurodevelopmental disorders, ADHD and ASD.

## 2. Materials and Methods

### 2.1. Patients

Patient data collection was conducted at the Sleep/Wake-Behavior Clinic in the Division of Child & Adolescent Psychiatry of BC Children’s Hospital in Vancouver, Canada. Patients were referred to the clinic by the hospital and/or community-based general practitioners, pediatricians, or psychiatrists. We conducted a retrospective analysis of the prospectively collected data via structured intake forms (Appendix A) and clinical assessments from our clinic over a time period of 18 months (2021 to 2023).

### 2.2. Inclusion Criteria

(1) Completion of electronic intake forms independently or with the assistance of the team prior to the clinical assessment. (2) Full clinical assessment (in-person or via telehealth) that included the suggested clinical immobilization test (SCIT) of the patient and/or accompanying biological family member (parent and/or sibling; note that applicability of the formal SCIT depends on the developmental age and capacity of the participating patient and the applicability of the informal SCIT on age and ability to move; see Figure 1). (3) Collection of information on the patient’s most recent iron status and parental ID history. Furthermore, we collected information on demographics, comorbidities and medications. Exclusion criteria were any hematologic comorbidities such as thalassemia or sickle cell anemia. Patients included had a mean age of 10.83 years (median: 12 years; min: 3 months; max: 23 years).

### 2.3. Iron Deficiency

Iron status was investigated in 199 patients. Non-anemic ID was defined as normal hemoglobin with serum ferritin < 50 µg/L, as per the RLS guidelines from the International Restless Legs Syndrome Study Group, with negative CRP and no signs of inflammation/infection in the lab (e.g., elevated CRP or WBC/differential) and/or clinically (e.g., asthma, eczema, acne, or parasites), and/or a fasting iron status with low fasting serum iron, low iron saturation, and/or high TIBC [19,24,25]. Additional RBC indices were reviewed as indicators of anemia, including mean corpuscular volume (MCV), mean corpuscular hemoglobin (MCH), and mean corpuscular hemoglobin concentration (MCHC). Parents with a clear self-reported history of ID were appropriately identified, and in cases where the evidence was inconclusive but clinical symptoms of restlessness and/or RLS were evident, parental serum ferritin values were reviewed using electronic records with the consent of the individual (no additional data collection). ID in adults was defined as serum ferritin < 75 µg/L with negative CRP and no clinical or laboratory signs of inflammation/infection [19].

### 2.4. Clinical Assessment

The screening and assessment methods for an in-depth assessment of disruptive sleep/wake-behaviors were developed during an interdisciplinary PhD research endeavor utilizing qualitative methodologies to optimize clinical best practices, as previously described [25,36].

For transparency, an overview of the clinical assessment methodology is described below (Table 1 and Figure 1) [36,37,38]. A table version of the electronic intake forms is shown in Appendix A. The intake process utilizes REDCap, an electronic data collection tool [39], since 2021. All patients receive an intake form consisting of qualitative open-ended questions regarding observed/experienced sleep situations utilizing the bedtime, excessive daytime sleepiness, awakenings, routines, snoring or sleep-disordered breathing, quality of sleep, and non-specific concerns domains [40]. Furthermore, individual goals [41] and concerns [36] of the family and, if applicable, of the patient were collected. Information regarding daytime functioning and medication information was also collected. For quantitative information, the following questionnaires were used: the Sleep Disturbance Scale for Children (SDSC) [42] and the ADHD Rating Scale-IV [43].

**Figure 1 nutrients-16-03064-f001:**
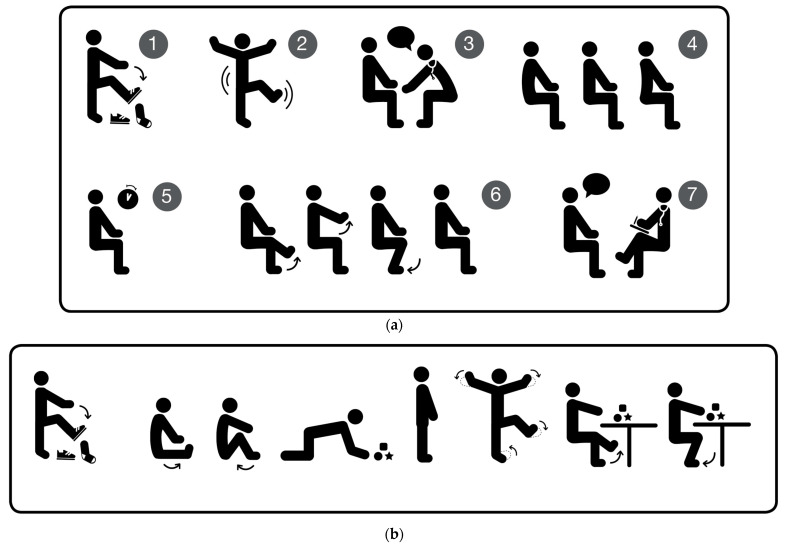
(**a**) Formal SCIT, visual depiction of the steps (①–⑦) as described in Table 1, which is used to capture descriptions of sensations from the affected individual in combination with observations from the assessing professional in a structured manner for exploring individual RLS symptoms (courtesy of kleanthes Publishers, Dresden [25,37,44]). (**b**) Informal SCIT, visual depiction of the play situations as described in Table 1, which is used in children and individuals who cannot participate in a formal SCIT. Here we explore the balance between movements and resting from an RLS perspective together with the caregivers based solely on observations (courtesy of kleanthes Publishers, Dresden [25,37,44]).

### 2.5. Data Analysis

We performed a comprehensive descriptive analysis to examine neurodevelopmental conditions, mental health diagnoses, sleep disorders, and medication use, stratified by ID status (ID versus no ID, and ID with versus without family history). Frequency and percentage are presented for all categorical variables. We then used unadjusted logistic regression to investigate the association of ID with family history (versus no ID or ID with no family history) and various sleep/wake disorders among the two main patient cohorts of ADHD and ASD. Finally, multivariate logistic regression models were constructed for the main sleep disorder, RLS (RLS only, familial RLS, and probable painful RLS) with ID (ID with a family history versus ID without family history versus no ID), ADHD, ASD, and the clinical RLS-associated symptoms of bedtime resistance, restlessness before falling asleep, and restlessness during sleep, as well as age and sex. Odds ratios (OR) with 95% confidence intervals (CI) and two-sided *p*-values are reported.

## 3. Results

### 3.1. Descriptive Statistics

Out of 250 patients referred to the Sleep/Wake-Behavior clinic between 2021 and 2023, 199/250 (80%) patients met the inclusion criteria. In total, 21 patients were excluded due to incomplete intake forms and another 30 patients were excluded because they had only been screened and not fully assessed at the time of analysis.

Based on the lab findings of the 199 included patients, 188 (94%) patients fulfilled the criteria for anemic or non-anemic ID, whereas 11 (6%) patients did not have biochemical evidence of ID. In total, 41% of all patients had a family history of ID. A family history of ID mainly traced back to birth mothers who experienced chronic or intermittent ID since their teenage years and/or during pregnancy. Fathers less frequently reported experiencing ID, but if so, patients had more first-degree relatives with ID. We further explored the history of ID and sleep in the patient’s siblings. Iron deficiency diagnosis for family members was typically made by their family doctors or pediatricians and was reported by the parent or caregiver during the screening and/or assessment.

Based on previous clinical assessments, 100 (50%) patients presented with an externalizing disorder or disorders of disruptive behaviors, with ADHD being the most common diagnosis (*n* = 92; 46%). In total, 84 (42%) patients had an internalizing disorder, with anxiety disorders being the most common (*n* = 81; 41%). The most common neurodevelopmental presentations were ASD (*n* = 89; 45%), neurologic conditions (*n* = 38, 19%), global developmental delay/intellectual disability (*n* = 26; 13%), and genetic conditions (*n* = 21; 11%).

Based on the clinical assessment at the Sleep/Wake-Behaviour Clinic, 121 (61%) patients met the *International Classification of Sleep Disorders* (Third Edition) criteria [45] for chronic insomnia and had problems with falling and/or maintaining sleep. In total, 32 (16%) patients fulfilled the criteria for circadian rhythm sleep disorder (CRSD); subtypes were captured, but as our cohort comprised individuals of various ages and changing sleep patterns, we are not presenting these data. In total, 43 (22%) patients experienced parasomnias, and 100 (50%) had signs of sleep-disordered breathing. In total, 148 (74%) patients had RLS, 103 (52%) had a family history of RLS, and 22 (11%) had probable painful RLS. Among the 22 patients with painful RLS, 9 (41%) had self-injurious behaviors; among this group, 6 (67%) had a family history of ID and 3 (33%) had no family history of ID. PLMS/Restless sleep was noted in 60 (30%) patients.

Descriptive statistics are presented in Table 2.

### 3.2. Subgroup Analysis ADHD

In sub-analyses of only patients with ADHD (*n* = 92), a family history of ID significantly increased the odds of having familial RLS (OR: 5.98, *p* = 0.0002, 95% CI [2.35, 15.20]), insomnia/DIMS (OR: 3.44, *p* = 0.0084, 95% CI [1.37, 8.64]), and RLS (OR: 7.00, *p* = 0.0014, 95% CI [1.49, 32.93]). There was also evidence of increased odds of probable painful RLS (OR: 3.48, *p* = 0.053, 95% CI [0.99, 12.30]) and SWTD (OR: 4.41, *p* = 0.079, 95% CI [0.84, 23.17]). The results of all of the odds ratio analysis are presented in Appendix A.

### 3.3. Subgroup Analysis ASD

In sub-analyses of patients with ASD (*n* = 89), a family history of ID significantly increased the odds of having insomnia/DIMS (OR: 4.77, *p* = 0.0014, 95% CI [1.82, 12.48]), RLS/PLMS/Restlessness (OR: 5.83, *p* = 0.0094, 95% CI [1.54, 22.07]), RLS (OR: 4.05, *p* = 0.013, 95% CI [1.33, 12.33]), and familial RLS (OR: 2.82, *p* = 0.02, 95% CI [1.17, 6.81]). The results of all odds ratios analysis are presented in Appendix A.

### 3.4. Multivariate Logistic Regression Analyses

Analysis of RLS only: A family history of ID was associated with over six times the odds of having RLS, compared to patients with no ID (OR: 6.25, *p* = 0.013, 95% CI [1.46, 26.74]), while no family history of ID was associated with over twice the odds, though this was not significant at *p* < 0.05 (OR: 2.67, *p* = 0.158, 95% CI [0.68, 10.39]). Patients with ADHD had about twice the odds of having RLS (OR: 2.12, *p* = 0.046, 95% CI [1.01, 4.42]) compared to those with no ADHD; ASD was not associated with RLS. Bedtime resistance and restlessness did not appear to be associated with RLS, either. See Appendix A.

Familial RLS analysis: A family history of ID was associated with over four times the odds of familial RLS compared to patients with no ID (OR: 4.38, *p* = 0.048, 95% CI [1.01, 18.94]). ADHD was not associated with RLS, but patients with ASD had about 70% higher odds of familial RLS, although this was not significant at *p* < 0.05 (OR: 1.71, *p* = 0.103, 95% CI [0.90, 3.24]). Bedtime resistance and restlessness did not appear to be associated with familial RLS. See Appendix A.

Probable Painful RLS analysis: Patients without ID were excluded from the analysis, as none of them had probable painful RLS. A family history of ID was associated with over twice the odds of probable painful RLS compared to patients with no family history of ID, although this was not significant at *p* < 0.05 (OR: 2.20, *p* = 0.113, 95% CI [0.83, 5.82]). ADHD and ASD were not associated with probable painful RLS, and neither were bedtime resistance or restlessness. See Appendix A.

### 3.5. Medications

Results are presented in Appendix A.

## 4. Discussion

This QIQA-study identified ID and family history of ID as being significant risk factors for sleep problems in patients referred to a Sleep/Wake-Behavior Clinic located in the Division of Child & Adolescent Psychiatry in a tertiary care hospital. As part of the project that included a structured intake procedure, all patients completed intake forms, had their iron status checked, and received clinical assessments. Notably, 94% of patients were diagnosed with either anemic or non-anemic ID. Among these patients, MH/NDDs, such as ADHD and ASD, along with sleep disorders such as RLS (76%) and insomnia (61%), were the most prevalent comorbidities. Prior scoping reviews conducted by our team have revealed associations between ID in children/adolescents and prevalent NDDs and MH conditions such as ADHD and ASD [3,6]. Upon reviewing the ADHD and ASD subgroups in our patient data, we identified specific sleep/wake-disorders that were more likely to occur in children/adolescents with a family history of ID. In the ADHD sub-group, patients with family history of ID had an increased risk of familial RLS, insomnia/DIMS, RLS, probable painful RLS, and sleep-wake transition disorder, compared to patients without ID or with no family history of ID. For patients with ASD, a family history of ID increased the risk of insomnia/DIMS, RLS/PLMS, RLS, and familial RLS compared to patients without ID or no family history of ID.

Interestingly, restlessness, a common symptom of sleep/wake disturbances that can be associated with ID [38], emerged as a strong risk factor across both the overall patient population in our review as well as the ADHD and ASD subgroups. Despite its high prevalence, restlessness is often missed in pediatric patients, likely due to several reasons. Firstly, pediatric sleep medicine has historically been derived from adult sleep medicine. Typically, pediatric patients are not actively involved in the clinical assessments and the symptom of restlessness is not explored from the child’s or adolescents’ perspective. In adult neurology, RLS is based on history-taking and the explicit mentioning of uncomfortable sensations. However, in pediatric sleep medicine, the proactive and structured involvement of pediatric patients in the assessment of their sleep/wake behaviors is a recent development [18,25,36]. Another reason restlessness is commonly missed is because the primary method of assessment, verbal communication, may pose challenges for pediatric patients who struggle to articulate their discomfort and pain, e.g., for young, non-verbal patients, and/or patients with developmental delays and intellectual disabilities [46]. Most importantly, restlessness is considered to be an intrinsic characteristic of patients with NDDs and MH-conditions such as ADHD, ASD, and PAE/FASD, and remains largely unexplored [2,4,29,47]. This is particularly evident in conditions such as RLS, where symptoms may be underrecognized due to the patients’ inability to express their discomfort accurately, or because they may not recognize the ‘baseline’ state [25,36]. Considering these factors, we aimed to ensure that restlessness was not overlooked. By following the recommendation of the Pediatric International RLS Study Group to conduct structured behavioral observations [18], we adapted the formal SCIT and applied an informal SCIT in cases where a formal SCIT was not applicable. The informal SCIT allows the observer and caregiver to review and explore movement patterns of the affected individual from an RLS perspective (see Table 1, Figure 1a,b). The methodology of explorative history-taking, combined with the review of a family history of RLS and ID among the patient’s parents and siblings helped to collect additional information in a structured manner and may have created a unique perspective for our patient population.

While the high percentage of pediatric patients diagnosed with RLS might be explained by exploratory observations, including the non-formal SCIT and investigation of family history, diagnosing probable painful RLS involves exploring its potential link with self-injurious behaviors (SIBs) in a high-risk population whose behaviors might be triggered by pain. Thus, in British Columbia, all pediatric patients referred to the Provincial SIB-Clinic receive a sleep/wake behavior assessment, and the frequency and intensity of the SIBs are explored using the explorative questioning criteria as described (Table 3). While SIBs in children with NDDs have been associated with frustrations in communication [48], which may lead to learned SIBs, painful RLS in children has not been described yet. Therefore, we have used the diagnosis of probable painful RLS retrospectively only in patients whose behavioral patterns and patterns of falling asleep/sleep maintenance and restlessness started to change after iron treatment. Our attempt to review restlessness by applying the formal and informal SCIT and conducting a structured iron status revealed an interesting new perspective. Out of the 22 patients with painful RLS, only 9 had self-injurious behaviors and 6/9 had a family history of ID, showing that in the subpopulation with self-injurious behaviors, ID, and particularly a family history of ID, may act as aggravating factors of self-injurious behaviors. The clinical conclusion is that the link between ID and SIBs should be further investigated.

## 5. Limitations and Strengths

(1) Reviewing a subgroup of patients all presenting with NDDs and MH conditions and not having a control group may have confounded clinical results. (2) Only a subgroup of patients were medication-naive, and some of the medications (e.g., stimulants or antipsychotics) may affect sensations associated with RLS and/or ADHD. For example, the negative association between sensory processing dysfunction (SPD) and a family history of ID in the ADHD subgroup is not explainable without the data of controls and without reviewing the effects of medications on sensory processing. (3) We have reported a retrospective analysis of data, despite collecting the data prospectively. A limitation in retrospectively analyzing data is the absence of a structured categorization for certain data which may have helped strengthen our analysis. For example, we did not distinguish between anemic and nonanemic ID in our electronic intake forms. Additionally, data regarding family history of ID, either of their mother or father, was not separated. Lastly, the relatively small sample sizes and small number of patients without ID reduced the statistical power in logistic regression analyses and resulted in reduced precision/wide confidence intervals. While all of these limitations raise questions about the generalizability of our results to the broader pediatric population, the high prevalence of ID (94%) and family history of ID (43%), and the potential link between self-injurious behaviors with ID serve as the strengths of our study. The mechanisms of ID in this population are likely multifactorial with inadequate nutrition, inflammation, and malabsorption being potential contributors. However, as our analysis was limited, we did not capture information on the possible causes. These aspects contribute valuable insights to the ongoing discourse about how ID and non-anemic ID should be explored [49].

## 6. Conclusions

While in-depth phenotyping of restlessness has been applied for some time in the field of sleep medicine [20,50], definitions of pediatric restlessness in sleep have been based on adult criteria and do not fully acknowledge physiological differences present in children. Therefore, further phenotyping of pediatric restlessness is necessary to identify these signs and symptoms in a clinical setting. In the context of ADHD, daytime restlessness has been considered to be a key diagnostic criterion, but nighttime restlessness and restless sleep are not included in the current DSM-V criteria [51], despite ADHD being a 24 h disorder. Interestingly, restless sleep was part of the diagnostic criteria for ADHD in the 1980 DSM-III, but has since been omitted in subsequent revisions [52,53]. The recent polysomnography-based descriptions for ADHD [31] and RSD [23] sleep phenotypes ask for a more in-depth investigation of clinical observations and biochemical imbalances, with ID and other factors affecting iron homeostasis (such as inflammation) becoming a focus of interest. To apply this concept, we developed an exploratory observation-based approach to RLS [25,36,44,54] and investigated ID among patients with NDDs and MH conditions. Our results indicate that ID and a family history of ID are significant risk factors for restless sleep and wake behaviors in patients with ADHD and ASD. This finding is critical, as ID can often be effectively treated with iron supplementation, potentially improving sleep in children and adolescents affected by these conditions.

## Figures and Tables

**Table 1 nutrients-16-03064-t001:** Suggested Clinical Immobilization Test procedure adapted from Ipsiroglu et al. [25,44].

Instructions for Patients and Accompanying Parents. * Ask child and accompanying parent/caregiver(s) to: *			
(①) Remove shoes/socks.	(②) Stand up, stretch, and shake out.	(③–⑤) Sit down on a height-wise-appropriate chair, barefoot, with feet flat on the floor and suppress any movements for approx. 2 min. (⑥) Lean back and take three slow, deep breaths. Now try to relax for approx. 2 min.	(⑦) Describe any sensations during each step of the process in any parts of the body. Sensations related to RLS can also occur in the arms or shoulders. When the test is carried out, participants are allowed to move as much as they want.
Instructions for Clinician. * Observe movement patterns and review the change in the movement patterns with the patient/caregiver: *			
(①) Provide clean sheets of paper or a mat for the floor to avoid the effects of confounding factors such as a cold or dirty floor, etc.	(②) Accompany the patient and their caregiver in standing up, stretching, shaking out, etc.; this reduces stress for the child/adolescent and improves the atmosphere.	(③–⑤) Observe the sitting position; patients usually sit under tension in order to suppress any movements and they may express which sitting positions help them become more relaxed. If this is helpful, continue for the entire 2 min. (⑥) Ask the child to now take three deep breaths (you can demonstrate that) during the exercise and observe the position of shoulders and comment (“still up” or “moving slowly down—is it difficult to relax?”). Usually, during the three breaths, patients begin to explain their sensations. Suggested monitoring time is 2 min.	(⑦) Observe any movement patterns, including twitches and small jerky movements during each step of the process and explore why they occurred and whether there were any associated sensations. Review both participants simultaneously and make comparisons; this allows them to reduce stress and make this a fun activity. Note: laughing supports relaxation, and during laughing, characteristic movements may happen. Observe the compensatory movements of the participants after the test is officially finished; suppressing movements usually ends in presenting with compensatory movements.
**General comments:** Create interactions, e.g., explain to the child that this is a game and that you are trying to understand who is able to sit longer without moving or ‘cheating’ (e.g., increasing tension, making slight movements). Observe the child’s ability to relax and/or increase tension. Make a joke to allow the child to relax and see whether there are any involuntary movements that occur during laughing. Usually, you need to repeat the test so that the child is familiar with the procedure. Ask the child and then the accompanying parent/caregiver(s) if they understand the procedure; try to create a collaborative discussion about words or phrases which describe how the child is feeling. Please do not suggest words to the child and remind them that their legs must be relaxed and not tense; you may check this manually by feeling extremities for any tension—do not forget to ask the child for permission before touching them.			
FORMAL SCIT: Follow instructions as described above and clinician mark down observed movement patterns and record location and described sensations below:			
*‘Described sensations’: ………*			
Sensations in toes/feet/legs		Has an urge to move, but unable to specify	
Sensations in fingers/hands/arms		Other (Specify):	
**No Sensations**			
Observations of movement:			
Difficulties sitting still		Twitching	
Increases tension in order to sit still		Other (Specify):	
**No observable movement patterns**			
INFORMAL SCIT: Observe the longest periods of rest. Choose a typical 5 min period during the assessment when the child is beginning to become more interactive (not shy or bored). Describe sequences of movement and rest patterns. Describe patterns the child applies in order to stay still/stop movements if the child is not able to participate in a formal SCIT (Figure 1a).			
*‘Described sensations”:*			
Historically described sensations relieving discomfort (example: leg massage, tight hugs, etc.)			
Parents’ descriptions (narrative):			
**No sensations**		**Not reported**	
Observations: Sitting position with increased tension			
At edge of chair		With legs swinging/kicking	
With legs/feet crossed		On lower legs/feet	
In abnormal positions (e.g., yogi-like positions)		Other (specify):	
Observations: Movement patterns			
Stretching/constant movement of toes/feet/legs		Rubbing toes/feet/legs or clenching to increase tension	
Repetitive movements of toes/feet/legs		Raising heels	
Stretching/constant movement of fingers/hands/arms		Rubbing fingers/hands/arms or clenching to increase tension	
Repetitive movement of fingers/hands/arms		Raising arms	
Other (Specify):			
**No observable movement patterns**			

**Table 2 nutrients-16-03064-t002:** Frequency and percent of patients with various comorbidities across different categories: all patients, patients without ID, patients with ID, patients without a family history of ID, and patients with a family history of ID.

	All Patients N = 199 Male/Female: 107/92	All Patients without ID N = 11 (6%) Male/Female: 7/4	All Patients with ID N = 188		
			All Patients with ID N = 188 (94%) Male/Female: 100/88	No Family History of ID N = 107 (57%) Male/Female: 50/57	Family History of ID N = 81 (43%) Male/Female: 50/31
Neurodevelopmental Conditions					
Autism spectrum disorder	**89 (44.7%)**	**2 (18.2%)**	**87 (46.3%)**	44 (41.1%)	43 (53.1%)
Fetal alcohol spectrum disorder	**9 (4.5%)**	**1 (9.1%)**	**8 (4.3%)**	4 (3.7%)	4 (4.9%)
Global developmental delay and intellectual disability	**26 (13.1%)**	**1 (9.1%)**	**25 (13.3%)**	15 (14.0%)	10 (12.3%)
Genetic conditions (Trisomy 21; Rett syndrome; Prader–Willi syndrome; Becker muscular dystrophy; Beckwith Wiedemann syndrome; Ehlers Danlos syndrome; CPT1 deficiency; DDX3X syndrome; 22q11.2 deletion syndrome (DiGeorge syndrome); Pierre Robin sequence; GLUT1 deficiency; gene mutations, micro-deletions, or abnormalities)	**21 (10.6%)**	**1 (9.1%)**	**20 (10.6%)**	15 (14.0%)	5 (6.2%)
Neurologic conditions (Cerebral palsy /hereditary spastic paraplegia; Leigh syndrome; Tourette syndrome, tics; epilepsy; visual/hearing impairment; mild traumatic brain injury; dysgraphia; chronic headaches)	**38 (19.1%)**	**4 (36.4%)**	**34 (18.1%)**	22 (20.6%)	12 (14.8%)
Sensory processing disorder	**17 (8.5%)**	**0 (0%)**	**17 (9.0%)**	12 (11.2%)	5 (6.2%)
Self-injurious behaviors	**26 (13.1%)**	**1 (9.1%)**	**25 (13.3%)**	12 (11.2%)	13 (16.0%)
Others (diabetes; hypothyroidism)	**2 (1.0%)**	**0 (0%)**	**2 (1.1%)**	2 (1.9%)	0 (0%)
**Mental health Diagnoses/Comorbidities**					
**Externalizing disorders**					
ADHD	**92 (46.2%)**	**4 (36.4%)**	**88 (46.8%)**	48 (44.9%)	40 (49.4%)
Oppositional defiant disorder	**6 (3.0%)**	**1 (9.1%)**	**5 (2.7%)**	4 (3.7%)	1 (1.2%)
Obsessive compulsive disorder	**20 (10.1%)**	**0 (0%)**	**20 (10.6%)**	10 (9.3%)	10 (12.3%)
**Internalizing disorders**					
Anxiety disorders	**81 (40.7%)**	**2 (18.2%)**	**79 (42.0%)**	44 (41.1%)	35 (43.2%)
Depression	**28 (14.1%)**	**1 (9.1%)**	**27 (14.4%)**	14 (13.1%)	13 (16.0%)
Depression with suicidal ideation	**5 (2.5%)**	**0 (0%)**	**5 (2.7%)**	3 (2.8%)	2 (2.5%)
Bipolar disorder	**3 (1.5%)**	**0 (0%)**	**3 (1.6%)**	1 (0.9%)	2 (2.5%)
**Sleep Disorders**					
Insomnia	**121 (60.8%)**	**7 (63.6%)**	**114 (60.6%)**	53 (49.5%)	61 (75.3%)
Excessive daytime sleepiness/disorders of excessive somnolence	**53 (26.6%)**	**5 (45.5%)**	**48 (25.5%)**	21 (19.6%)	27 (33.3%)
Circadian rhythm sleep disorder	**32 (16.1%)**	**5 (45.5%)**	**27 (14.4%)**	16 (15.0%)	11 (13.6%)
Parasomnias	**43 (21.6%)**	**2 (18.2%)**	**41 (21.8%)**	22 (20.6%)	19 (23.5%)
Sleep-disordered breathing	**100 (50.3%)**	**4 (36.4%)**	**96 (51.1%)**	57 (53.3%)	39 (48.1%)
Non-restorative sleep	**98 (49.2%)**	**2 (18.2%)**	**96 (51.1%)**	55 (51.4%)	41 (50.6%)
Sleep–wake transition disorders	**14 (7.0%)**	**1 (9.1%)**	**13 (6.9%)**	2 (1.9%)	11 (13.6%)
Periodic limb movements/Restless sleep	**60 (30.2%)**	**1 (9.1%)**	**59 (31.4%)**	32 (29.9%)	27 (33.3%)
Restless legs syndrome	**148 (74.4%)**	**5 (45.5%)**	**143 (76.1%)**	76 (71.0%)	67 (82.7%)
Familial restless legs syndrome	**103 (51.8%)**	**3 (27.3%)**	**100 (53.2%)**	44 (41.1%)	56 (69.1%)
Probable painful restless legs syndrome	**22 (11.1%)**	**0 (0%)**	**22 (11.7%)**	9 (8.4%)	13 (16.0%)

**Table 3 nutrients-16-03064-t003:** The five essential diagnostic criteria for RLS [20], and the adaptations, as well as examples, of pediatric patients. Note that in some patients, parental descriptions are used for describing the behavioral observations.

Essential Diagnostic Criteria for RLS (All Must Be Met):	Clinical Explorative Application of the Essential Diagnostic Criteria in Pediatric Patients	Examples in Children and Adolescents
An urge to move the legs usually, but not always, accompanied by, or felt to be caused by, uncomfortable and unpleasant sensations in the legs.	Description of fidgety behaviors	A seven-year-old boy, when asked to relax: “it is intense, I usually relax when I run”.
2. The urge to move the legs and any accompanying unpleasant sensations begin or worsen during periods of rest or inactivity such as lying down or sitting.	Favorite movement patterns: climbing, stretching, bumping toes.	A seven-year-old girl diagnosed with ADHD, when asked to relax: “Buzzing! Buzzzzzzing!!! My legs, my body are buzzzzzzzzzzzing!!!!”
	E.g., when a child’s leg movement is restricted, they become upset, but when given the freedom to move, their mood improves.	The mother of a one-year-old child: “Less resistance at bed time”.
	Bedtime resistance.	The mother of a two-year-old child: “Not being afraid to go to sleep”.
	Affected amount of sleep due to challenges in falling asleep and/or sleep maintenance.	The mother of a two-year-old: “My son is able to sleep the amount he is supposed to get at his age which is 12–13 h. In evenings, he is max getting 6 h on and off”.
	Hypermotor restlessness associated with sensory seeking behaviors with a focus on lower and/or upper limbs.	An eight-year-old non-verbal girl with ASD and ADHD diagnosed with painful RLS: “At night, she does not like feeling sleepy and has to sleep. She then jumps up, down, screams, and expresses SIB… …pulls at her pinky of fingers and toes throughout day (multiple times a day) or rams pinky into something hard. She pinches self and stomps toes on floor or rubs feet. She suddenly sits up and pulling at her pinky and gets up if she can to stomp feet into the ground. She pinches her mom when mom prevents SIB”. She kicks toes into floor causing problems walking. SIB can be associated with screaming when she is very distressed. SIB started age 3 and a half and started with rubbing feet. Then started jumping and slamming knees into ground. She pinches mom when mom prevents SIB. She always starts with crying. At night, she does not like feeling sleepy and has to sleep. She then jumps up, down, screams, and expresses SIB. She kicks toes into floor causing problems walking. SIB can be associated with screaming when she is very distressed. SIB started age 3 and a half and started with rubbing feet. Then started jumping and slamming knees into ground. She pinches mom when mom prevents SIB. She always starts with crying. At night, she does not like feeling sleepy and has to sleep. She then jumps up, down, screams, and expresses SIB”.
3. The urge to move the legs and any accompanying unpleasant sensations are partially or totally relieved by movement, such as walking or stretching, at least as long as the activity continues.	Favorite movement patterns: climbing, stretching, bumping toes, etc.	A 15-year-old boy, non-verbal with ASD and ADHD, developed his own nighttime routine as described by his parents: “… will run up and down the stairs climbing in and out of the bathtub, turning the water on while fully dressed in his PJs, until he feels content to finally retreat back to bed and try to settle for the night. If we try to help him or disrupt his “routines” it only escalates the behaviours. He doesn’t seem to tire, and will go on for an hour or more on nights when it’s really bad. Some school mornings he simply is too tried to attend school and wake up”.
	Unusual routines	The 14-year-old sister, speaking about her brother: “he runs up and down the stairs and when I ask him what he is doing, he says, he prepares himself for bed”.
4. The urge to move the legs and any accompanying unpleasant sensations during rest or inactivity only occur or are worse in the evening or night than during the day.	Restlessness before bedtime, behaviors, e.g., fidgeting at breakfast vs. dinner table.	The mother of a four-year-old boy: “Understand/treating the source of … crying”.
		The mother of a three-year-old female: “That we will rely on too many medications to help us fall asleep, and or stay asleep. And for myself to combat that drowsiness a few hours later with coffee, because I have to be up with my other two children”.
		The mother of the 15-year-old boy, non-verbal with ASD, ADHD: “During the night, his brothers have reported he will sometimes still be awake, humming, walking around, and or turning water on and off in the bathroom sinks and or flushing toilets…”
5. The occurrence of the above features is not solely accounted for as symptoms are primary to another medical or a behavioral condition (e.g., myalgia, venous stasis, leg edema, arthritis, leg cramps, positional discomfort, habitual foot tapping).	Criteria #5 makes quality control of the probable RLS-treatment strategy necessary; if the treatment with iron supplementation is successful, then RLS as a main diagnosis has to be considered.	

## Data Availability

The original contributions presented in the study are included in the article and Appendix A; further inquiries can be directed to the corresponding author.

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
