# Peer review of "Iron Deficiency and Restless Sleep/Wake Behaviors in Neurodevelopmental Disorders and Mental Health Conditions"

_nutrients, 2024, doi:10.3390/nu16183064_

Round 1

Reviewer 1 Report

Comments and Suggestions for Authors

The topic if the manuscript is important and interesting. The title of the manuscript reflect the content.

The abstract provides a good summary of the manuscript’s content.

Row 94: [29]and space between

Rows 94-95: Restless sleep disorder (RSD) – the abbreviation is already introduced earlier

Materials and methods: how was the family history of ID identified? (self-declaration or genetic mutation, etc.) It is an important aspect, because in the Results and Conclusion section, there are highlighted the correlations between the family history of ID and the different pathologies.

Row 111: past tense is recommended (was conducted)

Rows 128 and 138: regarding the serum ferritin level in the case of non-anemic ID, justification and reference is needed, because WHO defines low serum ferritin <12 μg/L for children and <15 μg/L for adults.

Table 2: abbreviation of EDS/DOES needs to be explained

Results: Graphical representation of odds ratios (instead of the tabulated format) would make the presentation of results much more transparent, and in this way could be included in the manuscript also.

Results and discussion: according to the data included in Supplementary Table 6a and 6b, there are patients on iron medications – in this case the start of beginning iron supplementation, the dose and mode of the administration are important data and cand influence the diagnosis of ID – these aspects should be discussed.

Row 331: abbreviation of WED needs to be explained

Overall comment: the connection between the iron status and restless leg syndrome is well known (doi: 10.1002/mdc3.12047), this aspect should be mentioned in the Introduction section and the Conclusion section – comparing the obtained results with other studies (https://doi.org/10.3389/fneur.2020.00298; doi: 10.1093/pch/17.4.193).

References are relevant to the study and properly cited.

Author Response

Comment 1: The topic if the manuscript is important and interesting. The title of the manuscript reflect the content.The abstract provides a good summary of the manuscript’s content.

Response 1: Thank you. 

Comment 2: Row 94: [29]and space between. 

Response 2: Thank you; space has been added. 

Comment 3: Rows 94-95: Restless sleep disorder (RSD) – the abbreviation is already introduced earlier. 

Response 3: Thank you, we changed this. 

Comment 4: Materials and methods: how was the family history of ID identified? (self-declaration or genetic mutation, etc.) It is an important aspect, because in the Results and Conclusion section, there are highlighted the correlations between the family history of ID and the different pathologies.

Response 4: Thank you. This piece is actually already included in the methods section on page 3.  “Parents with a clear self-reported history of ID were appropriately identified, and in cases where the evidence was inconclusive but clinical symptoms of restlessness and/or RLS were evident, parental serum ferritin values were reviewed using electronic records with the consent of the individual (no additional data collection).”

Comment 5: Row 111: past tense is recommended (was conducted). 

Response 5: Thank you, to make the manuscript more personal, we had used “we conducted”, we hope this is okay?

Comment 6: Rows 128 and 138: regarding the serum ferritin level in the case of non-anemic ID, justification and reference is needed, because WHO defines low serum ferritin <12 μg/L for children and <15 μg/L for adults.

Response 6: Thank you, we have referenced the International Restless Legs Syndrome Study Group RLS Guidelines.

"Iron status was investigated in 199 patients. Non-anemic ID was defined as normal hemoglobin with serum ferritin <50µg/L as per the RLS guidelines from the International Restless Legs Syndrome Study Group, with negative CRP and no signs of inflammation/infection in the lab (e.g., elevated CRP, WBC/differential) and/or clinically (e.g., asthma, eczema, acne, parasites), and/or a fasting iron status with low fasting serum iron, low iron saturation and/or high TIBC [18], [22], [23]."

Comment 7: Table 2: abbreviation of EDS/DOES needs to be explained

Response 7: Thank you, we have written out this abbreviation, among others within this table in order to make it understandable. 

Comment 8: Results: Graphical representation of odds ratios (instead of the tabulated format) would make the presentation of results much more transparent, and in this way could be included in the manuscript also.

Response 8: Thank you. We have included the graphical representation of odds ratios as supplementary figures (Supplementary Figures 1-3). 

Comment 9: Results and discussion: according to the data included in Supplementary Table 6a and 6b, there are patients on iron medications – in this case the start of beginning iron supplementation, the dose and mode of the administration are important data and cand influence the diagnosis of ID – these aspects should be discussed.

Response 9: Thank you, we agree that evaluation of the iron intervention would be important. However, this is beyond the scope of this paper and our current ethics board approval which is for a quality improvement quality assurance project. 

Comment 10: Row 331: abbreviation of WED needs to be explained

Response 10: Thank you. We have removed the term “WED”  to be consistent with the rest of the manuscript. 

Comment 11: Overall comment: the connection between the iron status and restless leg syndrome is well known (doi: 10.1002/mdc3.12047), this aspect should be mentioned in the Introduction section and the Conclusion section – comparing the obtained results with other studies (https://doi.org/10.3389/fneur.2020.00298; doi: 10.1093/pch/17.4.193).

Response 11: Thank you. We have incorporated these references. 

Comment 12: References are relevant to the study and properly cited.

Response 12: Thank you. 

Reviewer 2 Report

Comments and Suggestions for Authors

The paper entitled: 'Iron Deficiency and Restless Sleep/Wake-Behaviors in Neurodevelopmental Disorders and Mental Health Conditions" is well written and has high clinical importance.

The authors tried to understand the role of iron deficiency and Restless Sleep/Wake-Behaviors in mental conditions and neurodevelopmental disorders.

Major: In the discussion section authors need to describe the mechanisms of Iron Deficiency related to disorders mentioned above in more detail.  Are there any sex differences in this study? This information needs to be added in the "methods section" or "the limitations of the study".

Minor: Authors should remove a grey background from the tables.

Author Response

The paper entitled: 'Iron Deficiency and Restless Sleep/Wake-Behaviors in Neurodevelopmental Disorders and Mental Health Conditions" is well written and has high clinical importance.

The authors tried to understand the role of iron deficiency and Restless Sleep/Wake-Behaviors in mental conditions and neurodevelopmental disorders.

Comment 1: Major: In the discussion section authors need to describe the mechanisms of Iron Deficiency related to disorders mentioned above in more detail.  Are there any sex differences in this study? This information needs to be added in the "methods section" or "the limitations of the study".

Response 1: Thank you for your comments. For the mechanisms of iron deficiency, we have added this into the limitations section, as this was not analyzed in detail in the current study. 

“The mechanisms of ID in this population are likely multifactorial with inadequate nutrition, inflammation, and malabsorption being potential contributors. However, as our analysis was limited, we did not capture information on the possible causes.”

​Regarding sex, we have added the sex breakdown of patients in Table 2. For the multivariate logistic regression, these were also constructed for age and sex; this data is presented in the supplemental tables 2-4. 

Comment 2: Minor: Authors should remove a grey background from the tables.

Response 2: Thank you for your comment. In Table 1, we wanted to keep this grey colouring within the background of the table. This was developed with the Restless Legs Versus Growing Pains Study Group and we agreed on using shades of grey to make it printer-friendly and maintain the ability to differentiate between the various pieces of the table.

Round 2

Reviewer 2 Report

Comments and Suggestions for Authors

I recommend to accept the manuscript in its present form